# Dynamical mean-field theory for stochastic gradient descent in Gaussian mixture classification

**Francesca Mignacco**[1]    **Florent Krzakala**[2,3]    **Pierfrancesco Urbani**[1]

**Lenka Zdeborová**[1,4]

[1] Institut de physique théorique, Université Paris-Saclay, CNRS, CEA, Gif-sur-Yvette, France
[2] Laboratoire de Physique, CNRS, École Normale Supérieure, PSL University, Paris, France
[3] IdePHICS Laboratory, EPFL, Switzerland
[4] SPOC Laboratory, EPFL, Switzerland
Correspondence to: `francesca.mignacco@ipht.fr`

## Abstract

We analyze in a closed form the learning dynamics of stochastic gradient descent (SGD) for a single layer neural network classifying a high-dimensional Gaussian mixture where each cluster is assigned one of two labels. This problem provides a prototype of a non-convex loss landscape with interpolating regimes and a large generalization gap. We define a particular stochastic process for which SGD can be extended to a continuous-time limit that we call stochastic gradient flow. In the full-batch limit, we recover the standard gradient flow. We apply dynamical mean field theory from statistical physics to track the dynamics of the algorithm in the high-dimensional limit via a self-consistent stochastic process. We explore the performance of the algorithm as a function of control parameters shedding light on how it navigates the loss landscape.

## 1   Introduction

Understanding how stochastic gradient descent (SGD) manages to train artificial neural networks with good generalization capabilities by exploring the high-dimensional non-convex loss landscape is one of the central problems in theory of machine learning. A popular attempt to explain this behavior is by showing that the loss landscape itself is simple, with no spurious (i.e. leading to bad test error) local minima. Some empirical evidence instead leads to the conclusion that the loss landscape of state-of-the-art deep neural networks actually has spurious local (or even global) minima and stochastic gradient descent is able to find them [1, 2]. Still, the stochastic gradient descent algorithm, initialized at random, leads to good generalization properties in practice. It became clear that a theory that would explain this success needs to account for the whole trajectory of the algorithm. Yet this remains a challenging task, certainly for the state-of-the art deep networks trained on real datasets.

**Related work —**   A detailed description of the whole trajectory taken by the (stochastic) gradient descent was so far obtained only in several special cases. First such case are deep linear networks where the dynamics of gradient descent has been analyzed [3, 4]. While this line of works has led to very interesting insights about the dynamics, linear networks lack the expressivity of the non-linear ones and the large time behavior of the algorithm can be obtained with a simple spectral algorithm. Moreover, the analysis of dynamics in deep linear networks was not extended to the case of stochastic gradient descent. Second case where the trajectory of the algorithm was understood in detail is the *one-pass* (online) stochastic gradient descent for two-layer neural networks with a small

hidden layer in the teacher-student setting [5–9]. However, the one-pass assumption made in those analyses is far from what is done in practice and is unable to access the subtle difference between the training and test error that leads to many of the empirical mysteries observed in deep learning. A third very interesting line of research that recently provided insight about the behavior of stochastic gradient descent concerns two layer networks with divergingly wide hidden layer. This mean-field limit [10–12] maps the dynamics into the space of functions where its description is simpler and the dynamics can be written in terms of a closed set of differential equations. It is not clear yet whether this analysis can be extended in a sufficiently explicit way to deeper or finite width neural networks. The term *mean-field* has been used in several contexts in machine learning [13–18]. Note that the term in the aforementioned works refers to a variety of approximations and concepts. In this work we use it with the same meaning as in [19–21]. Most importantly, the term mean-field in our case has nothing to do with the width of an eventual hidden layer. We refer to [22] for a broader methodological review of mean-field methods and their applications to neural networks.

Our present work inscribes in the above line of research offering the dynamical mean-field theory (DMFT) formalism [19–21] leading to a closed set of integro-differential equations to track the full trajectory of the gradient descent (stochastic or not) from random initial condition in the high-dimensional limit for in-general non-convex losses. While in general the DMFT is a heuristic statistical physics method, it has been amenable to rigorous proof in some cases [23]. This is hence an important future direction for the case considered in the present paper. The DMFT has been applied recently to a high-dimensional inference problem in [24, 25] studying the spiked matrix-tensor model. However, this problem does not allow a natural way to study the stochastic gradient descent or to explore the difference between training and test errors. In particular, the spiked matrix-tensor model does not allow for the study of the so-called interpolating regime, where the loss function is optimized to zero while the test error remains positive. As such, its landscape is intrinsically different from supervised learning problems since in the former the spurious minima proliferate at high values of the loss while the good ones lie at the bottom of the landscape. Instead, deep networks have both spurious and good minima at 100% training accuracy and their landscape resembles much more the one of continuous constraint satisfaction problems [26, 27].

**Main contributions —** We study a natural problem of supervised classification where the input data come from a high-dimensional Gaussian mixture of several clusters, and all samples in one cluster are assigned to one of two possible output labels. We then consider a single-layer neural network classifier with a general non-convex loss function. We analyze a stochastic gradient descent algorithm in which, at each iteration, the batch used to compute the gradient of the loss is extracted at random, and we define a particular stochastic process for which SGD can be extended to a continuous time limit that we call stochastic gradient flow (SGF). In the full-batch limit we recover the standard Gradient Flow (GF). We describe the high-dimensional limit of the randomly initialized SGF with the DMFT that leads to a description of the dynamics in terms of a self-consistent stochastic process that we compare with numerical simulations. In particular, we show that the finite batch size can have a beneficial effect in the test error and acts as an effective regularization that prevents overfitting.

## 2   Setting and definitions

In all what follows, we will consider the high-dimensional setting where the dimension of each point in the dataset is $d \to \infty$ and the size of the training set $n = \alpha d$, being $\alpha$ a control parameter that we keep of order one. We consider a training set made of $n$ points

$$\mathbf{X} = (\mathbf{x}_1, ... \mathbf{x}_n)^\top \in \mathbb{R}^{n \times d} \quad \text{with labels} \quad \mathbf{y} = (y_1, ... y_n)^\top \in \{+1, -1\}^n \tag{1}$$

The patterns $\mathbf{x}_\mu$ are given by

$$\mathbf{x}_\mu = c_\mu \frac{\mathbf{v}^*}{\sqrt{d}} + \sqrt{\Delta} \, \mathbf{z}_\mu \qquad \mathbf{z}_\mu \sim \mathcal{N}(\mathbf{0}, \mathbf{I}_d) \quad \mu = 1, ... n \, . \tag{2}$$

Without loss of generality, we choose a basis where $\mathbf{v}^* = (1, 1, ... 1) \in \mathbb{R}^d$.

**Two-cluster dataset:** We will illustrate our results on a two-cluster example where the coefficients $c_\mu$ are taken at random $c_\mu = \pm 1$ with equal probability. Therefore one has two symmetric clouds of Gaussian points centered around two vectors $\mathbf{v}^*$ and $-\mathbf{v}^*$. The labels of the data points are fixed by

$y_\mu = c_\mu$. If the noise level $\Delta$ of the number of samples is small enough, the two Gaussian clouds are linearly separable by an hyperplane, as specified in detail in [28], and therefore a single layer neural network is enough to perform the classification task in this case. We hence consider learning with the simplest neural network that classifies the data according to $\hat{y}_\mu(\mathbf{w}) = \mathrm{sgn}[\mathbf{w}^\top \mathbf{x}_\mu / \sqrt{d}]$.

**Three-cluster dataset:** We consider also an example of three clusters where a good generalization error cannot be obtained by separating the points linearly. In this case we define $c_\mu = 0$ with probability $1/2$, and $c_\mu = \pm 1$ with probability $1/2$. The labels are then assigned as

$$y_\mu = -1 \text{ if } c_\mu = 0 \text{ , and } y_\mu = 1 \text{ if } c_\mu = \pm 1 \,. \tag{3}$$

One has hence three clouds of Gaussian points, two external and one centered in zero. In order to fit the data we consider a single layer-neural network with the door activation function, defined as

$$\hat{y}_\mu(\mathbf{w}) = \mathrm{sgn}\left[\left(\frac{\mathbf{w}^\top \mathbf{x}_\mu}{\sqrt{d}}\right)^2 - L^2\right] \,. \tag{4}$$

The onset parameter $L$ could be learned, but we will instead fix it to a constant.

**Loss function:** We study the dynamics of learning by the empirical risk minimization of the loss

$$\mathcal{H}(\mathbf{w}) = \sum_{\mu=1}^n \ell\left[y_\mu \phi\left(\frac{\mathbf{w}^\top \mathbf{x}_\mu}{\sqrt{d}}\right)\right] + \frac{\lambda}{2}\|\mathbf{w}\|_2^2, \tag{5}$$

where we added a Ridge regularization term. The activation function $\phi$ is given by

$$\phi(x) = \begin{cases} x & \text{linear for the two-cluster dataset} \\ x^2 - L^2 & \text{door for the three-cluster dataset} \,. \end{cases} \tag{6}$$

The DMFT analysis is valid for a generic loss function $\ell$. However, for concreteness in the result section we will focus on the logistic loss $\ell(v) = \ln\left(1 + e^{-v}\right)$. Note that in this setting the two-cluster dataset leads to convex optimization, with a unique minimum for finite $\lambda$, and implicit regularization for $\lambda = 0$ [29], and was analyzed in detail in [30, 28]. Still the performance of stochastic gradient descent with finite batch size cannot be obtained in *static* ways. The three-cluster dataset, instead, leads to a generically non-convex optimization problem which can present many spurious minima with different generalization abilities when the control parameters such as $\Delta$ and $\alpha$ are changed. We note that our analysis can be extended to neural networks with a small hidden layer [31], this would allow to study the role of overparametrization, but it is left for future work.

## 3 Stochastic gradient-descent training dynamics

**Discrete SGD dynamics —** We consider the discrete gradient-descent dynamics for which the weight update is given by

$$\mathrm{w}_j(t + \eta) = \mathrm{w}_j(t) - \eta\left[\lambda \mathrm{w}_j(t) + \sum_{\mu=1}^n s_\mu(t)\Lambda'\left(y_\mu, \frac{\mathbf{w}(t)^\top \mathbf{x}_\mu}{\sqrt{d}}\right)\frac{\mathrm{x}_{\mu,j}}{\sqrt{d}}\right] \tag{7}$$

where we have introduced the function $\Lambda(y, h) = \ell\left(y\phi\left(h\right)\right)$ and we have indicated with a prime the derivative with respect to $h$, $\Lambda'(y, h) = y\ell'\left(y\phi\left(h\right)\right)\phi'\left(h\right)$. We consider the following initialization of the weight vector $\mathbf{w}(0) \sim \mathcal{N}(\mathbf{0}, \mathbf{I}_d R)$, where $R > 0$ is a parameter that tunes the average length of the weight vector at the beginning of the dynamics[1]. The variables $s_\mu(t)$ are *i.i.d.* binary random variables. Their discrete-time dynamics can be chosen in two ways:

- In classical **SGD** at iteration $t$ one extracts the samples with the following probability distribution

$$s_\mu(t) = \begin{cases} 1 & \text{with probability } b \\ 0 & \text{with probability } 1 - b \end{cases} \tag{8}$$

  and $b \in (0, 1]$. In this way for each time iteration one extracts on average $B = bn$ patterns at random on which the gradient is computed and therefore the batch size is given by $B$. Note that if $b = 1$ one gets full-batch gradient descent.

- **Persistent-SGD** is defined by a stochastic process for $s_\mu(t)$ given by the following probability rules

$$\text{Prob}(s_\mu(t+\eta) = 1|s_\mu(t) = 0) = \frac{1}{\tau}\eta$$

$$\text{Prob}(s_\mu(t+\eta) = 0|s_\mu(t) = 1) = \frac{(1-b)}{b\tau}\eta,$$

(9)

where $s_\mu(0)$ is drawn from the probability distribution (8). In this case, for each time slice one has on average $B = bn$ patterns that are *active* and enter in the computation of the gradient. The main difference with respect to the usual SGD is that one keeps the same patterns and the same minibatch for a characteristic time $\tau$. Again, setting $b = 1$ one gets full-batch gradient descent.

**Stochastic gradient flow —** To write the DMFT we consider a continuous-time dynamics defined by the $\eta \to 0$ limit. This limit is not well defined for the usual SGD dynamics described by the rule (8) and we consider instead its *persistent* version described by Eq. (9). In this case the stochastic process for $s_\mu(t)$ is well defined for $\eta \to 0$ and one can write a continuous time equation as

$$\dot{\mathrm{w}}_j(t) = -\lambda\mathrm{w}_j(t) - \sum_{\mu=1}^{n} s_\mu(t)\Lambda'\left(y_\mu, \frac{\mathbf{w}(t)^\top\mathbf{x}_\mu}{\sqrt{d}}\right)\frac{\mathrm{x}_{\mu,j}}{\sqrt{d}},$$

(10)

Again, for $b = 1$ one recovers the gradient flow. We call Eq. (10) stochastic gradient flow (SGF).

## 4   Dynamical mean-field theory for SGF

We will now analyze the SGF in the infinite size limit $n \to \infty$, $d \to \infty$ with $\alpha = n/d$ and $b$ and $\tau$ fixed and of order one. In order to do that, we use dynamical mean-field theory (DMFT). The derivation of the DMFT equations is given in the supplementary material, here we will just present the main steps. The derivation extends the one reported in [32] for the non-convex perceptron model [26] (motivated there as a model of glassy phases of hard spheres). The main differences of the present work with respect to [32] are that here we consider a finite-batch gradient descent and that our dataset is structured while in [32] the derivation was done for full-batch gradient descent and random i.i.d. inputs and i.i.d. labels, i.e. a case where one cannot investigate generalization error and its properties. The starting point of the DMFT is the dynamical partition function

$$Z_{\text{dyn}} = \int_{\mathbf{w}(0)=\mathbf{w}^{(0)}} \mathcal{D}\mathbf{w}(t) \prod_{j=1}^{d} \delta\left[-\dot{\mathrm{w}}_j(t) - \lambda\mathrm{w}_j(t) - \sum_{\mu=1}^{n} s_\mu(t)\Lambda'\left(y_\mu, \frac{\mathbf{w}(t)^\top\mathbf{x}_\mu}{\sqrt{d}}\right)\frac{\mathrm{x}_{\mu,j}}{\sqrt{d}}\right], \quad (11)$$

where $\mathcal{D}\mathbf{w}(t)$ stands for the measure over the dynamical trajectories starting from $\mathbf{w}(0)$. Since $Z_{\text{dyn}} = 1$ (it is just an integral of a Dirac delta function) [33] one can average directly $Z_{\text{dyn}}$ over the training set, the initial condition and the stochastic processes of $s_\mu(t)$. We indicate this average with the brackets $\langle\cdot\rangle$. Hence we can write

$$Z_{\text{dyn}} = \left\langle \int \mathcal{D}\mathbf{w}(t)\mathcal{D}\hat{\mathbf{w}}(t)\, e^{S_{\text{dyn}}} \right\rangle,$$

(12)

where we have defined

$$S_{\text{dyn}} = \sum_{j=1}^{d} \int_0^{+\infty} \mathrm{d}t\, i\hat{\mathrm{w}}_j(t)\left(-\dot{\mathrm{w}}_j(t) - \lambda\mathrm{w}_j(t) - \sum_{\mu=1}^{n} s_\mu(t)\Lambda'\left(y_\mu, \frac{\mathbf{w}(t)^\top\mathbf{x}_\mu}{\sqrt{d}}\right)\frac{\mathrm{x}_{\mu,j}}{\sqrt{d}}\right). \quad (13)$$

and we have introduced a set of fields $\hat{\mathbf{w}}(t)$ to produce the integral representation of the Dirac delta function. The average over the training set can be then performed explicitly, and the dynamical partition function $Z_{\text{dyn}}$ is expressed as an integral of an exponential with extensive exponent in $d$:

$$Z_{\text{dyn}} = \int \mathcal{D}\mathbf{Q}\,\mathcal{D}\mathbf{m}\, e^{dS(\mathbf{Q},\mathbf{m})},$$

(14)

where $\mathbf{Q}$ and $\mathbf{m}$ are two dynamical order parameters defined in the supplementary material. Therefore, the dynamics in the $d \to \infty$ limit satisfies a large deviation principle and we can approximate $Z_{\text{dyn}}$

with its value at the saddle point of the action $S$. In particular, one can show that the saddle point equations for the parameters $\mathbf{Q}$ and $\mathbf{m}$ can be recast into a self consistent stochastic process for a variable $h(t)$ related to the typical behavior of $\mathbf{w}(t)^\top \mathbf{z}_\mu/\sqrt{d}$, which evolves according to the stochastic equation:

$$\partial_t h(t) = -(\lambda + \hat{\lambda}(t))h(t) - \sqrt{\Delta}s(t)\Lambda'(y(c), r(t) - Y(t)) + \int_0^t dt' M_R(t, t')h(t') + \xi(t), \quad (15)$$

where we have denoted by $r(t) = \sqrt{\Delta}h(t) + m(t)(c + \sqrt{\Delta}h_0)$ and $m(t)$ is the *magnetization*, namely $m(t) = \mathbf{w}(t)^\top \mathbf{v}^*/d$. The details of the computation are provided in the supplementary material. There are several sources of stochasticity in Eq. (15). First, one has a dynamical noise $\xi(t)$ that is Gaussian distributed and characterized by the correlations

$$\langle \xi(t) \rangle = 0, \qquad \langle \xi(t)\xi(t') \rangle = M_C(t, t'). \quad (16)$$

Furthermore, the starting point $h(0)$ of the stochastic process is random and distributed according to

$$P(h(0)) = e^{-h(0)^2/(2R)}/\sqrt{2\pi R}. \quad (17)$$

Moreover, one has to introduce a quenched Gaussian random variable $h_0$ with mean zero and average one. We recall that the random variable $c = \pm 1$ with equal probability in the two-cluster model, while $c = 0, \pm 1$ in the three-cluster one. The variable $y(c)$ is therefore $y(c) = c$ in the two-cluster case, and is given by Eq. (3) in the three-cluster one. Finally, one has a dynamical stochastic process $s(t)$ whose statistical properties are specified in Eq. (9). The magnetization $m(t)$ is obtained from the following deterministic differential equation

$$\partial_t m(t) = -\lambda m(t) - \mu(t), \qquad m(0) = 0^+. \quad (18)$$

The stochastic process for $h(t)$, the evolution of $m(t)$, as well as the statistical properties of the dynamical noise $\xi(t)$ depend on a series of kernels that must be computed self consistently and are given by

$$\begin{aligned}
\hat{\lambda}(t) &= \alpha\Delta \langle s(t)\Lambda''(y(c), r(t)) \rangle, \\
\mu(t) &= \alpha \left\langle s(t)\left(c + \sqrt{\Delta}h_0\right)\Lambda'(y(c), r(t)) \right\rangle, \\
M_C(t, t') &= \alpha\Delta \langle s(t)s(t')\Lambda'(y(c), r(t))\Lambda'(y(c), r(t')) \rangle, \\
M_R(t, t') &= \alpha\Delta \frac{\delta}{\delta Y(t')} \langle s(t)\Lambda'(y(c), r(t)) \rangle \Big|_{Y=0}.
\end{aligned} \quad (19)$$

In Eq. (19) the brackets denote the average over all the sources of stochasticity in the self-consistent stochastic process. Therefore one needs to solve the stochastic process in a self-consistent way. Note that $Y(t)$ in Eq. (15) is set to zero and we need it only to define the kernel $M_R(t, t')$. The set of Eqs. (15), (18) and (19) can be solved by a simple straightforward iterative algorithm. One starts with a guess for the kernels and then runs the stochastic process for $h(t)$ several times to update the kernels. The iteration is stopped when a desired precision on the kernels is reached [34].

Note that, in order to solve Eqs. (15), (18) and (19), one needs to discretize time. In the result section 5, in order to compare with numerical simulations, we will take the time-discretization of DMFT equal to the learning rate in the simulations. In the time-discretized DMFT, this allows us to extract the variables $s(t)$ either from (8) (SGD) or (9) (Persistent-SGD). In the former case this gives us a SGD-inspired discretization of the DMFT equations.

Finally, once the self-consistent stochastic process is solved, one has access also to the dynamical correlation functions $C(t, t') = \mathbf{w}(t) \cdot \mathbf{w}(t')/d$, encoded in the dynamical order parameter $\mathbf{Q}$ that appears in the large deviation principle of Eq. (14). $C(t, t')$ concentrates for $d \to \infty$ and therefore is

controlled by the equations

$$\partial_t C(t', t) = -\tilde{\lambda}(t) C(t, t') + \int_0^t \mathrm{d}s \, M_R(t, s) C(t', s) + \int_0^{t'} \mathrm{d}s \, M_C(t, s) R(t', s)$$

$$- m(t') \left( \int_0^t \mathrm{d}s M_R(t, s) m(s) + \mu(t) - \hat{\lambda}(t) m(t) \right) \quad \text{if } t \neq t',$$

$$\frac{1}{2} \partial_t C(t, t) = -\tilde{\lambda}(t) C(t, t) + \int_0^t \mathrm{d}s \, M_R(t, s) C(t, s) + \int_0^t \mathrm{d}s \, M_C(t, s) R(t, s) \qquad (20)$$

$$- m(t) \left( \int_0^t \mathrm{d}s \, M_R(t, s) m(s) + \mu(t) - \hat{\lambda}(t) m(t) \right),$$

$$\partial_t R(t, t') = -\tilde{\lambda}(t) R(t, t') + \delta(t - t') + \int_{t'}^t \mathrm{d}s \, M_R(t, s) R(s, t'),$$

where we used the shorthand notation $\tilde{\lambda}(t) = \lambda + \hat{\lambda}(t)$. We consider the linear response regime, and $R(t, t') = \sum_i \delta w_i(t) / \delta H_i(t') / d$ is a response function that controls the variations of the weights when their dynamical evolution is affected by an infinitesimal local field $H_i(t)$. Coupling a local field $H_i(t)$ to each variable $w_i(t)$ changes the loss function as follows: $\mathcal{H}(\mathbf{w}(t)) \to \mathcal{H}(\mathbf{w}(t)) - \sum_{i=1}^d H_i(t) w_i(t)$, resulting in an extra term $H_i(t)$ to the right hand side of Eq. (10). We then consider the limit $H_i(t) \to 0$. It is interesting to note that the second of Eqs. (20) controls the evolution of the norm of the weight vector $C(t, t)$ and even if we set $\lambda = 0$ we get that it contains an effective regularization $\hat{\lambda}(t)$ that is dynamically self-generated [35].

**Dynamics of the loss and the generalization error —** Once the solution for the self-consistent stochastic process is found, one can get several interesting quantities. First, one can look at the training loss, which can be obtained as

$$e(t) = \alpha \langle \Lambda(y, r(t)) \rangle, \qquad (21)$$

where again the brackets denote the average over the realization of the stochastic process in Eq. (15). The training accuracy is given by

$$a(t) = 1 - \langle \theta(-y\phi(r(t))) \rangle \qquad (22)$$

and, by definition, it is equal to one as soon as all vectors in the training set are correctly classified. Finally, one can compute the generalization error. At any time step, it is defined as the fraction of mislabeled instances:

$$\varepsilon_{\text{gen}}(t) = \frac{1}{4} \mathbb{E}_{\mathbf{X}, \mathbf{y}, \mathbf{x}_{\text{new}}, y_{\text{new}}} \left[ (y_{\text{new}} - \hat{y}_{\text{new}}(\mathbf{w}(t)))^2 \right], \qquad (23)$$

where $\{\mathbf{X}, \mathbf{y}\}$ is the training set, $\mathbf{x}_{\text{new}}$ is an unseen data point and $\hat{y}_{\text{new}}$ is the estimator for the new label $y_{\text{new}}$. The dependence on the training set here is hidden in the weight vector $\mathbf{w}(t) = \mathbf{w}(t, \mathbf{X}, \mathbf{y})$. In the two-cluster case one can easily show that

$$\varepsilon_{\text{gen}}(t) = \frac{1}{2} \text{erfc} \left( \frac{m(t)}{\sqrt{2\Delta \, C(t, t)}} \right). \qquad (24)$$

Conversely, for the door activation trained on the three-cluster dataset we get

$$\varepsilon_{\text{gen}}(t) = \frac{1}{2} \text{erfc} \left( \frac{L}{\sqrt{2\Delta C(t, t)}} \right) + \frac{1}{4} \left( \text{erf} \left( \frac{L - m(t)}{\sqrt{2\Delta C(t, t)}} \right) + \text{erf} \left( \frac{L + m(t)}{\sqrt{2\Delta C(t, t)}} \right) \right). \qquad (25)$$

## 5   Results

In this section, we compare the theoretical curves resulting from the solution of the DMFT equations derived in Sec. 4 to numerical simulations. This analysis allows to gain insight into the learning dynamics of stochastic gradient descent and its dependence on the various control parameters in the two models under consideration.

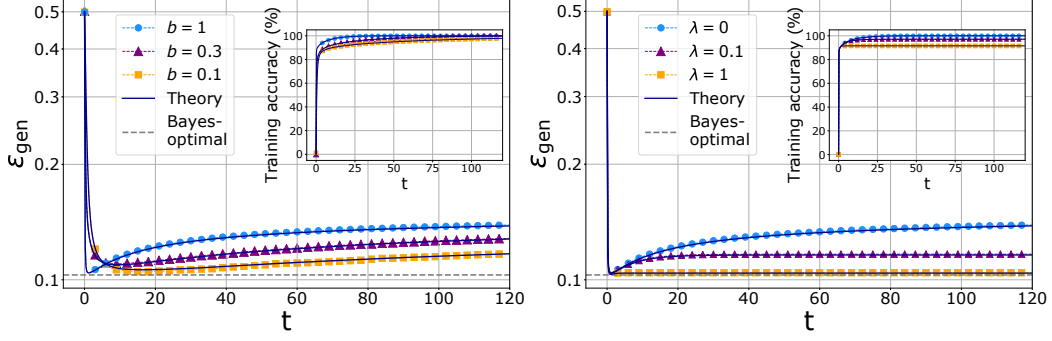

Figure 1: **Left:** Generalization error as a function of the training time for Persistent-SGD in the two-cluster model, with $\alpha = 2$, $\Delta = 0.5$, $\lambda = 0$, $1/\tau = 0.6$ and different batch sizes $b = 1, 0.3, 0.1$. The continuous lines mark the numerical solution of DMFT equations, while the symbols are the results of simulations at $d = 500$, $\eta = 0.2$, and $R = 0.01$. The dashed grey line marks the Bayes-optimal error from [28]. **Right:** Generalization error as a function of the training time for full-batch gradient descent in the two-cluster model with different regularization $\lambda = 0, 0.1, 1$ and the same parameters as in the left panel. In each panel, the inset shows the training accuracy as a function of the training time.

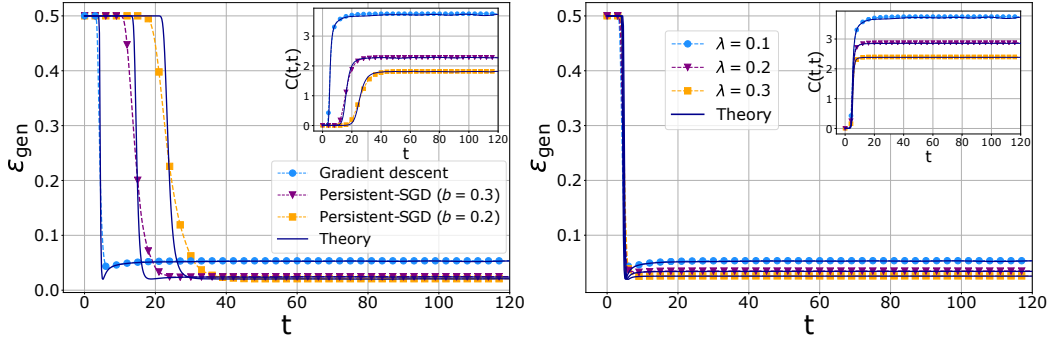

Figure 2: **Left:** Generalization error as a function of the training time in the three-cluster model, at fixed $\alpha = 3$, $\Delta = 0.05$, $L = 0.7$, $\lambda = 0.1$, for full-batch gradient descent and Persistent-SGD with different batch size $b = 0.2, 0.3$ and activation rate $1/\tau = b$. The continuous lines mark the numerical solution of DMFT equations, the symbols represent simulations at $\eta = 0.2$, $R = 0.01$, and $d = 5000$. **Right:** Generalization error as a function of training time for full-batch gradient descent in the three-cluster model, at fixed $\alpha = 3$, $\Delta = 0.05$, $L = 0.7$, $\eta = 0.2$, $R = 0.01$, and different regularization $\lambda = 0.1, 0.2, 0.3$. The simulations are done at $d = 5000$. In each panel, the inset shows the norm of the weights as a function of the training time.

The left panel of Fig. 1 shows the learning dynamics of the Persistent-SGD in the two-cluster model without regularization $\lambda = 0$. We clearly see a good match between the numerical simulations and the theoretical curves obtained from DMFT, notably also for small values of batchsize $b$ and dimension $d = 500$. The figure shows that there exist regions in control parameter space where Persistent-SGD is able to reach 100% training accuracy, while the generalization error is bounded away from zero. Remarkably, we observe that the additional noise introduced by decreasing the batch size $b$ results in a shift of the early-stopping minimum of the generalization error at larger times and that, on the time window we show, a batch size smaller than one has a beneficial effect on the generalization error at long times. The right panel illustrates the role of regularization in the same model trained with full-batch gradient descent, presenting that regularization has a similar influence on the learning curve as small batch-size but without the slow-down incurred by Persistent-SGD.

The influence of the batch size $b$ and the regularization $\lambda$ for the three-cluster model is shown in Fig. 2. We see an analogous effect as for the two-clusters in Fig. 1. In the inset of Fig. 2, we show the norm of the weights as a function of the training time. Both with the smaller mini-batch size and larger regularization the norm is small, testifying further that the two play a similar role in this case.

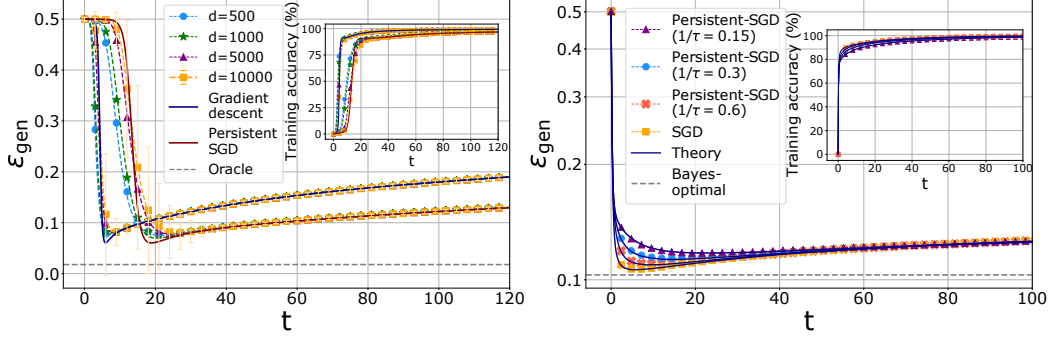

Figure 3: **Left:** Generalization error as a function of the training time for full-batch gradient descent and Persistent-SGD with $1/\tau = b = 0.3$ in the three-cluster model, at fixed $\alpha = 2$, $\Delta = 0.05$, $L = 0.7$ and $\lambda = 0$. The continuous lines mark the numerical solution of DMFT equations, the symbols represent simulations at $\eta = 0.2$, $R = 1$, and increasing dimension $d = 500, 1000, 5000, 10000$. Error bars are plotted for $d = 10000$. The dashed lines mark the oracle error (see supplementary material). **Right:** Generalization error as a function of the training time for Persistent-SGD with different activation rates $1/\tau = 0.15, 0.3, 0.6$ and classical SGD in the two-cluster model, both with $b = 0.3$, $\alpha = 2$, $\Delta = 0.5$, $\lambda = 0$, $\eta = 0.2$, $R = 0.01$. The continuous lines mark the numerical solution of DMFT equations (in case of SGD we use the SGD-inspired discretization), while the symbols represent simulations at $d = 500$. The dashed lines mark the Bayes-optimal error from [28]. In each panel, the inset displays the training accuracy as a function of time.

One difference between the two-cluster an the three-cluster models we observe concerns the behavior of the generalization error at small times. Actually, for the three-cluster model, good generalization is reached because of finite-size effects. Indeed, the corresponding loss function displays a $\mathbb{Z}_2$ symmetry according to which for each local minimum $\mathbf{w}$ there is another one $-\mathbf{w}$ with exactly the same properties. Note that this symmetry is inherited from the activation function $\phi$ (6), which is even. This implies that if $d \to \infty$, the generalization error would not move away from $0.5$ in finite time. However, when $d$ is large but finite, at time $t = 0$ the weight vector has a finite projection on $\mathbf{v}^*$ which is responsible for the dynamical symmetry breaking and eventually for a low generalization error at long times. In order to obtain an agreement between the theory and simulations, we initialize $m(t)$ in the DMFT equations with its corresponding finite-$d$ average value at $t = 0$. In the left panel of Fig. 3, we show that while this produces a small discrepancy at intermediate times that diminishes with growing size, at long times the DMFT tracks perfectly the evolution of the algorithm.

The right panel of Fig. 3 summarizes the effect of the characteristic time $\tau$ in the Persistent-SGD, i.e. the typical persistence time of each pattern in the training mini-batch. When $\tau$ decreases, the Persistent-SGD algorithm is observed to be getting a better early-stopping generalization error and the dynamics gets closer to the usual SGD dynamics. As expected, the $\tau \to 0$ limit of the Persistent-SGD converges to the SGD. It is remarkable that the SGD-inspired discretization of the DMTF equations, that is in principle an ad-hoc construction as the corresponding flow-limit in which the derivation holds does not exist, shows a perfect agreement with the numerics.

Fig. 4 presents the influence of the weight norm at initialization $R$ on the dynamics, for the two-cluster (left) and three-cluster (right) model. For the two-cluster case, the gradient descent algorithm with all-zeros initialization "jumps" on the Bayes-optimal error at the first iteration as derived in [28], and in this particular setting the generalization error is monotonically increasing in time. As $R$ increases the early stopping error gets worse. At large times all the initializations converge to the same value of the error, as they must, since this is a full-batch gradient descent without regularization that at large times converges to the max-margin estimator according to [29]. For the three-cluster model we observe a qualitatively similar behavior.

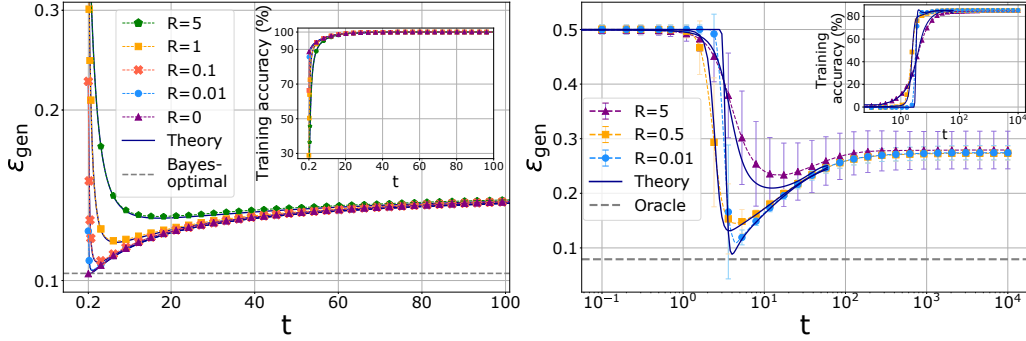

Figure 4: **Left:** Generalization error as a function of training time for full-batch gradient descent in the two-cluster model, at fixed $\alpha = 2$, $\Delta = 0.5$, $\lambda = 0$, $\eta = 0.2$, and different initialization variances $R = 0, 0.01, 0.1, 1, 5$. The continuous lines mark the numerical solution of DMFT equations, while the symbols represent simulations at $d = 500$. The dashed lines mark the Bayes-optimal error from [28]. The $y-$axis is cut for better visibility. **Right:** Generalization error as a function of training time for full-batch gradient descent in the three-cluster model, at fixed $\alpha = 3$, $\Delta = 0.1$, $\lambda = 0$, $\eta = 0.1$ and different initialization variances $R = 0.01, 0.5, 5$. The continuous lines mark the numerical solution of DMFT equations, while the symbols represent simulations at $d = 1000$. The dashed grey line marks the oracle error (see supplementary material). In each panel, the inset shows the training accuracy as a function of time.

## Broader Impact

Our work is theoretical in nature, and as such the potential societal consequence are difficult to foresee. We anticipate that deeper theoretical understanding of the functioning of machine learning systems will lead to their improvement in the long term.

## Acknowledgments and Disclosure of Funding

This work was supported by "Investissements d'Avenir" LabExPALM (ANR-10-LABX-0039-PALM), the ERC under the European Union's Horizon 2020 Research and Innovation Program 714608-SMiLe, as well as by the French Agence Nationale de la Recherche under grant ANR-17-CE23-0023-01 PAIL and ANR-19-P3IA-0001 PRAIRIE.

## Footnotes

[1]The DMFT equations we derive can be easily generalized to the case in which the initial distribution over $\mathbf{w}$ is different. We only need it to be separable and independent of the dataset.

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
