[Supplementary Material]

# Dynamical mean-field theory for stochastic gradient descent in Gaussian mixture classification – supplementary material

**Francesca Mignacco**[1]          **Florent Krzakala**[2,3]          **Pierfrancesco Urbani**[1]

**Lenka Zdeborová**[1,4]

[1] Institut de physique théorique, Université Paris-Saclay, CNRS, CEA, Gif-sur-Yvette, France
[2] Laboratoire de Physique, CNRS, École Normale Supérieure, PSL University, Paris, France
[3] IdePHICS Laboratory, EPFL, Switzerland
[4] SPOC Laboratory, EPFL, Switzerland
Correspondence to: `francesca.mignacco@ipht.fr`

## A   Derivation of the dynamical mean-field equations

The derivation of the self-consistent stochastic process discussed in the main text can be obtained using tools of statistical physics of disordered systems. In particular, it has been done very recently for a related model, the spherical perceptron with random labels, in [1]. Our derivation extends the known DMFT equations by including

- structure in the data;
- a stochastic version of gradient descent as discussed in the main text;
- the relaxation of the spherical constraint over the weights and the introduction of a Ridge regularization term.

There are at least two ways to write the DMFT equations. One is by using field theoretical techniques; otherwise one can employ a dynamical version of the so-called *cavity method* [2]. Here we opt for the first option that is generically very compact and immediate and it has a form that resembles very much a *static* treatment of the Gibbs measure of the problem [3]. We use a supersymmetric (SUSY) representation to derive the dynamical mean-field (DMFT) equations [1, 4]. We do not report all the details, that can be found in [1] along with an alternative derivation based on the cavity method, but we limit ourselves to provide the main points. We first consider the dynamical partition function, corresponding to Eq. (11) in the main text

$$
\begin{aligned}
Z_{\mathrm{dyn}} = \Bigg\langle \int \left[ \frac{\mathrm{d}\mathbf{w}^{(0)}}{(2\pi)^{\frac{d}{2}}} e^{-\frac{1}{2}\|\mathbf{w}^{(0)}\|_2^2} \right] \int_{\mathbf{w}(0)=\mathbf{w}^{(0)}} \mathcal{D}\mathbf{w}(t) \\
\times \prod_{j=1}^{d} \delta \left[ -\dot{\mathrm{w}}_j(t) - \lambda \mathrm{w}_j(t) - \sum_{\mu=1}^{n} s_\mu(t) \Lambda' \left( y_\mu, \frac{\mathbf{w}(t)^\top \mathbf{x}_\mu}{\sqrt{d}} \right) \frac{\mathrm{x}_{\mu,j}}{\sqrt{d}} \right] \Bigg\rangle,
\end{aligned}
\tag{A.1}
$$

where the brackets $\langle \cdot \rangle$ stand for the average over $s_\mu(t)$, $y_\mu$ and the realization of the noise in the training set. The average over the initial condition is written explicitly. Note that we choose an initial condition that is Gaussian, but we could have chosen a different probability measure over the initial configuration of the weights. The equations can be generalized to other initial conditions as soon as they do not depend on quenched random variables that enter in the stochastic gradient

descent (SGD) dynamics and their distribution is separable. As observed in the main text, we have that $Z_{\text{dyn}} = \langle Z_{\text{dyn}} \rangle = 1$. We can write the integral representation of the Dirac delta function in Eq. A.1 by introducing a set of fields $\hat{\mathbf{w}}(t)$

$$Z_{\text{dyn}} = \left\langle \int \mathcal{D}\mathbf{w}(t) \mathcal{D}\hat{\mathbf{w}}(t) \, e^{S_{\text{dyn}}} \right\rangle, \tag{A.2}$$

where the dynamical action $S_{\text{dyn}}$ is defined as in Eq. (13) of the main text

$$S_{\text{dyn}} = \sum_{j=1}^{d} \int_0^{+\infty} \mathrm{d}t \, i\hat{\mathrm{w}}_j(t) \left( -\dot{\mathrm{w}}_j(t) - \lambda \mathrm{w}_j(t) - \sum_{\mu=1}^{n} s_\mu(t) \Lambda' \left( y_\mu, \frac{\mathbf{w}(t)^\top \mathbf{x}_\mu}{\sqrt{d}} \right) \frac{\mathrm{x}_{\mu,j}}{\sqrt{d}} \right). \tag{A.3}$$

## A.1 SUSY formulation

The dynamical action $S_{\text{dyn}}$ (A.3) can be rewritten in a supersymmetric form, by extending the time coordinate to include two Grassman coordinates $\theta$ and $\bar{\theta}$, i.e. $t_a \to a = (t_a, \theta_a, \bar{\theta}_a)$. The dynamic variable $\mathbf{w}(t_a)$ and the auxiliary variable $i\hat{\mathbf{w}}(t_a)$ are encoded in a super-field

$$\mathbf{w}(a) = \mathbf{w}(t_a) + i\,\theta_a\bar{\theta}_a\hat{\mathbf{w}}(t_a). \tag{A.4}$$

From the properties of Grassman variables [5]

$$\theta^2 = \bar{\theta}^2 = \theta\bar{\theta} + \bar{\theta}\theta = 0,$$

$$\int \mathrm{d}\theta = \int \mathrm{d}\bar{\theta} = 0, \qquad \int \mathrm{d}\theta\,\theta = \int \mathrm{d}\bar{\theta}\,\bar{\theta} = 1, \tag{A.5}$$

$$\partial_\theta g(\theta) = \int \mathrm{d}\theta\, g(\theta) \quad \text{for a generic function } g,$$

it follows that

$$\int \mathrm{d}a\, f\left(\mathbf{w}(a)\right) = \int_0^{+\infty} \mathrm{d}t_a\, i\hat{\mathbf{w}}(t_a) f'\left(\mathbf{w}(t_a)\right). \tag{A.6}$$

We can use Eq. (A.6) to rewrite $S_{\text{dyn}}$. We obtain

$$S_{\text{dyn}} = -\frac{1}{2} \int \mathrm{d}a\mathrm{d}b\, \mathcal{K}(a,b)\mathbf{w}(a)^\top\mathbf{w}(b) - \sum_{\mu=1}^{n} \int \mathrm{d}a\, s_\mu(a)\, \Lambda\left(y_\mu, h_\mu(a)\right), \tag{A.7}$$

where we have defined $h_\mu(a) \equiv \mathbf{w}(a)^\top \mathbf{x}_\mu/\sqrt{d}$ and we have implicitly defined the kernel $\mathcal{K}(a,b)$ such that

$$-\frac{1}{2} \int \mathrm{d}a\mathrm{d}b\, \mathcal{K}(a,b)\mathbf{w}(a)^\top\mathbf{w}(b) = \sum_{j=1}^{d} \int_0^{+\infty} \mathrm{d}t\, i\hat{\mathrm{w}}_j(t)\left( -\dot{\mathrm{w}}_j(t) - \lambda \mathrm{w}_j(t) \right). \tag{A.8}$$

By inserting the definition of $h_\mu(a)$ in the partition function, we have

$$Z_{\text{dyn}} = \left\langle \int \mathcal{D}\mathbf{w}(a)\mathcal{D}h_\mu(a)\mathcal{D}\hat{h}_\mu(a) \, \exp\left[ -\frac{1}{2} \int \mathrm{d}a\mathrm{d}b\, \mathcal{K}(a,b)\mathbf{w}(a)^\top\mathbf{w}(b) \right. \right.$$
$$\left. \left. - \sum_{\mu=1}^{n} \int \mathrm{d}a\, s_\mu(a)\, \Lambda\left(y_\mu, h_\mu(a)\right) \right] \exp\left[ \sum_{\mu=1}^{n} \int \mathrm{d}a\, i\,\hat{h}_\mu(a)\left( h_\mu(a) - \frac{\mathbf{w}(a)^\top \mathbf{x}_\mu}{\sqrt{d}} \right) \right] \right\rangle. \tag{A.9}$$

Let us consider the last factor in the integral in (A.9). We can perform the average over the random vectors $\mathbf{z}_\mu \sim \mathcal{N}(\mathbf{0}, \mathbf{I}_d)$, denoted by an overline, as

$$\overline{\exp\left[ \sum_{\mu=1}^{n} \int \mathrm{d}a\, i\,\hat{h}_\mu(a)\left( h_\mu(a) - \frac{\mathbf{w}(a)^\top \mathbf{x}_\mu}{\sqrt{d}} \right) \right]}$$

$$= \exp\left[ \sum_{\mu=1}^{n} \int \mathrm{d}a\, i\,\hat{h}_\mu(a)\left( h_\mu(a) - c_\mu m(a) - \sqrt{\frac{\Delta}{d}}\mathbf{w}(a)^\top \mathbf{z}_\mu \right) \right] \tag{A.10}$$

$$= \exp\left[ \sum_{\mu=1}^{n} \int \mathrm{d}a\, i\,\hat{h}_\mu(a)\left( h_\mu(a) - c_\mu m(a) \right) - \frac{\Delta}{2} \sum_{\mu=1}^{n} \int \mathrm{d}a\, \mathrm{d}b\, Q(a,b)\hat{h}_\mu(a)\hat{h}_\mu(b) \right],$$

where we have defined

$$
m(a) = \frac{1}{d}\mathbf{w}(a)^\top \mathbf{v}^*,
$$
$$
Q(a,b) = \frac{1}{d}\mathbf{w}(a)^\top \mathbf{w}(b).
$$

(A.11)

By inserting the definitions of $m(a)$ and $Q(a,b)$ in the partition function, we obtain

$$
Z_{\mathrm{dyn}} = \int \mathcal{D}\mathbf{Q}\,\mathcal{D}\mathbf{m}\; e^{dS(\mathbf{Q},\mathbf{m})},
$$

(A.12)

where $\mathbf{Q} = \{Q(a,b)\}_{a,b}$, $\mathbf{m} = \{m(a)\}_a$ and

$$
S(\mathbf{Q},\mathbf{m}) = \frac{1}{2}\log\det\left(Q(a,b) - m(a)m(b)\right) - \frac{1}{2}\int \mathrm{d}a\mathrm{d}b\, \mathcal{K}(a,b)Q(a,b) + \alpha \log \mathcal{Z},
$$
$$
\mathcal{Z} = \left\langle \int \mathcal{D}h(a)\mathcal{D}\hat{h}(a)\,\exp\left[-\frac{\Delta}{2}\int \mathrm{d}a\mathrm{d}b\, Q(a,b)\hat{h}(a)\hat{h}(b)\right.\right.
$$
$$
\left.\left. + \int \mathrm{d}a\, i\hat{h}(a)\left(h(a) - cm(a)\right) - \int \mathrm{d}a\, s(a)\,\Lambda\left(y,h(a)\right)\right]\right\rangle.
$$

(A.13)

We have used that the samples are i.i.d. and removed the index $\mu = 1,...n$. The brackets denote the average over the random variable $c$, that has the same distribution as the $c_\mu$, over $y$, distributed as $y_\mu$, and over the random process of $s(t)$, defined by Eq. (9) in the main text. If we perform the change of variable $Q(a,b) \leftarrow Q(a,b) + m(a)m(b)$, we obtain

$$
S(\mathbf{Q},\mathbf{m}) = \frac{1}{2}\log\det Q(a,b) - \frac{1}{2}\int \mathrm{d}a\mathrm{d}b\, \mathcal{K}(a,b)\left(Q(a,b) + m(a)m(b)\right) + \alpha \log \mathcal{Z},
$$
$$
\mathcal{Z} = \left\langle \int \mathcal{D}h(a)\mathcal{D}\hat{h}(a)\, e^{S_{\mathrm{loc}}}\right\rangle,
$$

(A.14)

where the effective local action $S_{\mathrm{loc}}$ is given by

$$
S_{\mathrm{loc}} = -\frac{\Delta}{2}\int \mathrm{d}a\mathrm{d}b\, Q(a,b)\hat{h}(a)\hat{h}(b) - \frac{\Delta}{2}\left(\int \mathrm{d}a\,\hat{h}(a)m(a)\right)^2
$$
$$
+ \int \mathrm{d}a\, i\hat{h}(a)\left(h(a) - cm(a)\right) - \int \mathrm{d}a\, s(a)\,\Lambda\left(y,h(a)\right).
$$

(A.15)

Performing a Hubbard-Stratonovich transformation on $\exp\left[-\frac{\Delta}{2}\left(\int \mathrm{d}a\,\hat{h}(a)m(a)\right)^2\right]$ and a set of transformations on the fields $h(a)$, we obtain that we can rewrite $\mathcal{Z}$ as

$$
\mathcal{Z} = \left\langle \int \frac{\mathrm{d}h_0}{\sqrt{2\pi}}e^{-\frac{h_0^2}{2}}\int \mathcal{D}h(a)\mathcal{D}\hat{h}(a)\,\exp\left[-\frac{1}{2}\int \mathrm{d}a\mathrm{d}b\, Q(a,b)\hat{h}(a)\hat{h}(b)\right.\right.
$$
$$
\left.\left. + \int \mathrm{d}a\, i\hat{h}(a)h(a) - \int \mathrm{d}a\, s(a)\,\Lambda\left(y,\sqrt{\Delta}h(a) + m(a)(c + \sqrt{\Delta}h_0)\right)\right]\right\rangle.
$$

(A.16)

## A.2 Saddle-point equations

We are interested in the large $d$ limit of $Z_{\mathrm{dyn}}$, in which, according to Eq. (A.12), the partition function is dominated by the saddle-point value of $S(\mathbf{Q},\mathbf{m})$:

$$
\begin{cases}
\left.\dfrac{\delta S(\mathbf{Q},\mathbf{m})}{\delta Q(a,b)}\right|_{(\mathbf{Q},\mathbf{m})=(\tilde{\mathbf{Q}},\tilde{\mathbf{m}})} = 0 \\[4mm]
\left.\dfrac{\delta S(\mathbf{Q},\mathbf{m})}{\delta m(a)}\right|_{(\mathbf{Q},\mathbf{m})=(\tilde{\mathbf{Q}},\tilde{\mathbf{m}})} = 0
\end{cases}.
$$

(A.17)

$\tilde{Q}(a,b)$ is obtained from the equation

$$
-\mathcal{K}(a,b) + Q^{-1}(a,b) + \frac{2\alpha}{\mathcal{Z}}\frac{\delta\mathcal{Z}}{\delta Q(a,b)} = 0.
$$

(A.18)

The saddle-point equation for $\tilde{m}(a)$ is instead

$$-\int db \, \mathcal{K}(a,b)m(b) + \frac{\alpha}{\mathcal{Z}}\frac{\delta \mathcal{Z}}{\delta m(a)} = 0. \tag{A.19}$$

It can be easily shown by exploiting the Grassmann structure of Eqs. (A.18)-(A.19) that they lead to a self consistent stochastic process described by

$$\dot{h}(t) = -\tilde{\lambda}(t)h(t) - \sqrt{\Delta}s(t)\Lambda'\left(y, r(t) - Y(t)\right) + \int_0^t dt' M_R(t,t')h(t') + \xi(t), \tag{A.20}$$

where the initial condition is drawn from $P(h(0)) \sim e^{-h(0)^2/(2R)}/\sqrt{2\pi}$, and $r(t) = \sqrt{\Delta}h(t) + m(t)(c + \sqrt{\Delta}h_0)$, with $P_0(h_0) \sim e^{-h_0^2/2}/\sqrt{2\pi}$. We have defined the auxiliary functions

$$\begin{aligned}
\mu(t) &= \alpha \left\langle s(t)\left(c + \sqrt{\Delta}h_0\right)\Lambda'\left(y, r(t)\right)\right\rangle \\
\hat{\lambda}(t) &= \alpha\Delta \left\langle s(t)\Lambda''\left(y, r(t)\right)\right\rangle \\
\tilde{\lambda}(t) &= \lambda + \hat{\lambda}(t)
\end{aligned} \tag{A.21}$$

and kernels

$$\begin{aligned}
M_C(t,t') &= \alpha\Delta \left\langle s(t)s(t')\Lambda'\left(y, r(t)\right)\Lambda'\left(y, r(t')\right)\right\rangle, \\
M_R(t,t') &= \alpha\Delta^{3/2} \left\langle s(t)s(t')\Lambda'\left(y, r(t)\right)\Lambda''\left(y, r(t')\right)i\hat{h}(t')\right\rangle \\
&\equiv \alpha\Delta \frac{\delta}{\delta Y(t')}\langle s(t)\Lambda'(y, r(t))\rangle\Big|_{Y=0}.
\end{aligned} \tag{A.22}$$

In addition, from (A.19), one can derive an ordinary differential equation for the magnetization

$$\dot{m}(t) = -\lambda m(t) - \mu(t). \tag{A.23}$$

The brackets in the previous equations denote, at the same time, the average over the label $y$, the process $s(t)$, as well as the average over the noise $\xi(t)$ and both $h_0$ and $h(0)$, whose probability distributions are given by $P(h(0))$ and $P_0(h_0)$ respectively. In other words, one has a set of kernels, such as $M_R(t,t')$ and $M_C(t,t')$, that can be obtained as average over the stochastic process for $h(t)$ and therefore must be computed self-consistently.

Finally, Eq. (A.18) gives rise to Eq. (20) of the main text while Eq. (A.19) gives rise to the equation for the evolution of the magnetization. Note that the norm of the weight vector $\mathbf{w}(t)$ can be also computed by sampling the stochastic process

$$\begin{aligned}
\dot{\mathbf{w}}(t) &= -\tilde{\lambda}(t)\mathbf{w}(t) + \int_0^t dt' M_R(t,t')(\mathbf{w}(t') - m(t')h_0) + \xi(t) + h_0(\hat{\lambda}(t)m(t) - \mu(t)), \\
P(\mathbf{w}_0) &= \frac{1}{\sqrt{2\pi R}}e^{-\mathbf{w}_0^2/(2R)},
\end{aligned} \tag{A.24}$$

from which one gets

$$C(t,t') = \langle \mathbf{w}(t)^2 \rangle. \tag{A.25}$$

### A.3 Numerical solution of DMFT equations

The algorithm to solve the DMFT equations that are summed up in Eq. (A.20) is the most natural one. It can be understood in this way. The outcome of the DMFT is the computation of the kernels and functions appearing in it, namely $m(t)$, $M_C(t,t')$ and so on. They are determined as averages over the stochastic process that is defined through them. Therefore, one needs to solve the system of equations in a self-consistent way. The straightforward way to do that is to proceed by iterations:

1. We start from a random guess of the kernels, that we use to sample the stochastic process (A.20) several times;

2. We compute the averages over these multiple realizations to obtain the updates of the auxiliary functions (A.21) and kernels (A.22), along with the magnetization (A.23);

3. We use these new guesses to sample again multiple realizations of the stochastic process;

4. We repeat steps 2. and 3. until the kernels reach a fixed point.

As in all iterative solutions of fixed point equations, it is natural to introduce some damping in the update of the kernels to avoid wild oscillations. Note that the DMFT fixed point equations are deterministic, hence at given initial condition the solution is unique. Indeed, the kernels computed by DMFT are causal and a simple integration scheme of the equations is just extending them progressively in time starting from their initial value, which is completely deterministic given the initial condition for the stochastic process. This procedure has been first implemented in [7, 8] and recently developed further in other applications [9, 10]. However, DMFT has a long tradition in condensed matter physics [11] where more involved algorithms have been developed.

# B  Generalization error

The generalization error at any time step is defined as the fraction of mislabeled instances:

$$\varepsilon_{\text{gen}}(t) \equiv \frac{1}{4}\mathbb{E}_{\mathbf{X},\mathbf{y},\mathbf{x}_{\text{new}},y_{\text{new}}}\left[\left(y_{\text{new}} - \hat{y}_{\text{new}}\left(\mathbf{w}(t)\right)\right)^2\right], \tag{B.1}$$

where $\{\mathbf{X}, \mathbf{y}\}$ is the training set, $\mathbf{x}_{\text{new}}$ is an unseen data point and $\hat{y}_{\text{new}}$ is the estimator for the new label $y_{\text{new}}$. The dependence on the training set here is hidden in the weight vector $\mathbf{w}(t) = \mathbf{w}(t, \mathbf{X}, \mathbf{y})$.

## B.1  Perceptron with linear activation function

In this case, the estimator for a new label is $\hat{y}_{\text{new}}\left(\mathbf{w}(t)\right) = \text{sign}\left(\mathbf{w}(t)^\top \mathbf{x}_{\text{new}}\right)$. The generalization error in the infinite dimensional limit $d \to \infty$ has been computed in [6] and reads

$$\varepsilon_{\text{gen}}(t) = \frac{1}{2}\text{erfc}\left(\frac{m(t)}{\sqrt{2\Delta\, C(t,t)}}\right). \tag{B.2}$$

## B.2  Perceptron with door activation function

In this case, the estimator for a new label is $\hat{y}_{\text{new}}\left(\mathbf{w}(t)\right) = \text{sign}\left(\frac{1}{d}(\mathbf{w}(t)^\top \mathbf{x}_{\text{new}})^2 - L^2\right)$. From Eq. (B.1), we have that

$$\varepsilon_{\text{gen}}(t) = \frac{1}{2}\left(1 - \mathbb{E}_{\mathbf{X},\mathbf{y},\mathbf{x}_{\text{new}},y_{\text{new}}}\left[y_{\text{new}} \cdot \hat{y}_{\text{new}}(\mathbf{w}(t))\right]\right). \tag{B.3}$$

We consider the second term of (B.3)

$$\mathbb{E}_{\mathbf{X},\mathbf{y},\mathbf{x}_{\text{new}},y_{\text{new}}}\left[y_{\text{new}} \cdot \hat{y}_{\text{new}}(\mathbf{w}(t))\right] = \mathbb{E}_{\mathbf{X},\mathbf{y},\mathbf{x}_{\text{new}}}\left[\text{sign}\left(\frac{y_{\text{new}}}{d}(\mathbf{w}(t)^\top \mathbf{x}_{\text{new}})^2 - y_{\text{new}}L^2\right)\right]. \tag{B.4}$$

In the high dimensional limit, the overlap between weight vector and data point at each time step concentrates

$$\frac{\mathbf{w}(t)^\top \mathbf{x}_{\text{new}}}{\sqrt{d}} = \frac{\mathbf{w}(t)^\top}{\sqrt{d}}\left(c_{\text{new}}\frac{\mathbf{v}^*}{\sqrt{d}} + \sqrt{\Delta}\,\mathbf{z}_{\text{new}}\right) \underset{d\to\infty}{\to} c_{\text{new}}\, m(t) + \sqrt{\Delta C(t,t)}\, z, \tag{B.5}$$

where $z \sim \mathcal{N}(0,1)$. Therefore, we obtain

$$\mathbb{E}_{\mathbf{X},\mathbf{y},\mathbf{x}_{\text{new}},y_{\text{new}}}\left[y_{\text{new}} \cdot \hat{y}_{\text{new}}(\mathbf{w}(t))\right] \simeq$$

$$\simeq \mathbb{E}_{c_{\text{new}},z,y_{\text{new}}}\left[\text{sign}\left(y_{\text{new}}\left(c_{\text{new}}\, m(t) + \sqrt{\Delta C(t,t)}\, z\right)^2 - y_{\text{new}}L^2\right)\right]$$

$$= \mathbb{P}\left(y_{\text{new}}\left(c_{\text{new}}\, m(t) + \sqrt{\Delta C(t,t)}\, z\right)^2 \geq y_{\text{new}}L^2\right) \tag{B.6}$$

$$-\mathbb{P}\left(y_{\text{new}}\left(c_{\text{new}}\, m(t) + \sqrt{\Delta C(t,t)}\, z\right)^2 < y_{\text{new}}L^2\right)$$

and the generalization error in the infinite dimensional limit $d \to \infty$ is

$$\varepsilon_{\text{gen}}(t) = (1-\rho)\text{erfc}\left(\frac{L}{\sqrt{2\Delta C(t,t)}}\right) + \frac{\rho}{2}\left(\text{erf}\left(\frac{L-m(t)}{\sqrt{2\Delta C(t,t)}}\right) + \text{erf}\left(\frac{L+m(t)}{\sqrt{2\Delta C(t,t)}}\right)\right). \quad \text{(B.7)}$$

## C  Oracle error

We call *oracle error* the classification error made by an ideal oracle that has access to the vector $\mathbf{v}^*$ that characterizes the centers of the clusters in the two models under consideration (see Sec. 2 in the main text). We define the oracle's estimator $\hat{y}_{\text{new}}^O$ given a new data point $\mathbf{x}_{\text{new}}$ as

$$\hat{y}_{\text{new}}^O = \arg\max_{\tilde{y}_{\text{new}}} \text{p}\left(\tilde{y}_{\text{new}}|\mathbf{x}_{\text{new}}\right), \quad \text{(C.1)}$$

where the prior over the label $\tilde{y}_{\text{new}}$ and the coefficient $\tilde{c}_{\text{new}}$ along with the channel distribution

$$\text{p}\left(\mathbf{x}_{\text{new}}|\tilde{c}_{\text{new}}\right) \propto \exp\left[-\frac{1}{2\Delta}\|\mathbf{x}_{\text{new}} - \frac{\tilde{c}_{\text{new}}}{\sqrt{d}}\mathbf{v}^*\|_2^2\right] \quad \text{(C.2)}$$

are known. We can rewrite the probability in Eq. (C.1) as

$$\text{p}\left(\tilde{y}_{\text{new}}|\mathbf{x}_{\text{new}}\right) \propto \sum_{\tilde{c}_{\text{new}}=0,\pm 1} \text{p}\left(\tilde{y}_{\text{new}}, \tilde{c}_{\text{new}}\right)\text{p}\left(\mathbf{x}_{\text{new}}|\tilde{c}_{\text{new}}\right) = (1-\rho)\delta(\tilde{y}_{\text{new}}+1)e^{-\frac{1}{2\Delta}\|\mathbf{x}_{\text{new}}\|_2^2}$$

$$+ \frac{\rho}{2}\delta(\tilde{y}_{\text{new}}-1)\left(e^{-\frac{1}{2\Delta}\|\mathbf{x}_{\text{new}}-\frac{1}{\sqrt{d}}\mathbf{v}^*\|_2^2} + e^{-\frac{1}{2\Delta}\|\mathbf{x}_{\text{new}}+\frac{1}{\sqrt{d}}\mathbf{v}^*\|_2^2}\right) \quad \text{(C.3)}$$

$$= e^{-\frac{1}{2\Delta}\|\mathbf{x}_{\text{new}}\|_2^2}\left[(1-\rho)\delta(\tilde{y}_{\text{new}}+1) + \rho\delta(\tilde{y}_{\text{new}}-1)e^{-\frac{1}{2\Delta}}\cosh\left(\frac{1}{\Delta\sqrt{d}}\mathbf{x}_{\text{new}}^\top\mathbf{v}^*\right)\right].$$

The oracle error is then

$$\varepsilon_{\text{gen}}^O = \mathbb{P}\left(\hat{y}_{\text{new}}^O \neq y_{\text{new}}\right) = (1-\rho)\,\mathbb{P}\left(\hat{y}_{\text{new}}^O = 1|y_{\text{new}}=-1\right) + \rho\,\mathbb{P}\left(\hat{y}_{\text{new}}^O = -1|y_{\text{new}}=1\right). \quad \text{(C.4)}$$

We can compute the two terms in the above equation separately

$$\mathbb{P}\left(\hat{y}_{\text{new}}^O = 1|y_{\text{new}}=-1\right) = \mathbb{P}\left(\rho e^{-\frac{1}{2\Delta}}\cosh\left(\frac{1}{\sqrt{\Delta d}}\mathbf{z}_{\text{new}}^\top\mathbf{v}^*\right) > 1-\rho\right)$$

$$= \mathbb{P}\left(\rho e^{-\frac{1}{2\Delta}}\cosh\left(\frac{\zeta_{\text{new}}}{\sqrt{\Delta}}\right) > 1-\rho\right) = \text{erfc}\left(\sqrt{\frac{\Delta}{2}}\left|\text{arccosh}\left(\frac{(1-\rho)}{\rho}e^{1/2\Delta}\right)\right|\right), \quad \text{(C.5)}$$

and

$$\mathbb{P}\left(\hat{y}_{\text{new}}^O = -1|y_{\text{new}}=1\right) = \mathbb{P}\left(1-\rho > \rho e^{-\frac{1}{2\Delta}}\cosh\left(\frac{c_{\text{new}}}{\Delta} + \frac{1}{\sqrt{\Delta d}}\mathbf{z}_{\text{new}}^\top\mathbf{v}^*\right)\right)$$

$$= \mathbb{P}\left(1-\rho > \rho e^{-\frac{1}{2\Delta}}\cosh\left(\frac{c_{\text{new}}}{\Delta} + \frac{\zeta_{\text{new}}}{\sqrt{\Delta}}\right)\right)$$

$$= \frac{1}{2}\left[\text{erf}\left(\frac{\Delta\left|\text{arccosh}\left(\frac{(1-\rho)}{\rho}e^{1/2\Delta}\right)\right|+1}{\sqrt{2\Delta}}\right) + \text{erf}\left(\frac{\Delta\left|\text{arccosh}\left(\frac{(1-\rho)}{\rho}e^{1/2\Delta}\right)\right|-1}{\sqrt{2\Delta}}\right)\right], \quad \text{(C.6)}$$

where $\mathbf{z}_{\text{new}} \sim \mathcal{N}(\mathbf{0}, \mathbf{I}_d)$, $\zeta_{\text{new}} \sim \mathcal{N}(0,1)$, and $c_{\text{new}} = \pm 1$ with probability $\frac{1}{2}$.
Finally, we obtain that the oracle error is

$$\varepsilon_{\text{gen}}^{BO} = (1-\rho)\text{erfc}\left(\sqrt{\frac{\Delta}{2}}\left|\text{arccosh}\left(\frac{(1-\rho)}{\rho}e^{1/2\Delta}\right)\right|\right)$$

$$+ \frac{\rho}{2}\left[\text{erf}\left(\frac{\Delta\left|\text{arccosh}\left(\frac{(1-\rho)}{\rho}e^{1/2\Delta}\right)\right|+1}{\sqrt{2\Delta}}\right) + \text{erf}\left(\frac{\Delta\left|\text{arccosh}\left(\frac{(1-\rho)}{\rho}e^{1/2\Delta}\right)\right|-1}{\sqrt{2\Delta}}\right)\right]. \quad \text{(C.7)}$$