[Reviews · NeurIPS 2020]

Review 1

Summary and Contributions: The authors apply dynamical mean-field theory (widely known in Physics) to study the learning dynamics of stochastic gradient descent in simple single-layer neural networks classifying a high-dimensional mixture of 2 or 3 Gaussians. Along the conventional SGD approach, the authors also introduce and study an alternative persistent-SGD sampling method that permits the continuous limit of the gradient descent dynamics. The authors derive dynamical mean-field theory equations for SGD and analyze these equations numerically showing a good agreement with direct simulations and exploring the parameter space of the problem. I thank the authors for their rebuttal. Having read it, I am more convinced that this work may have curious practical implications and may lead to other interesting results.

Strengths: Most of the paper claims are sound and are well supported both theoretically and empirically. Good agreement between theoretical and experimental results is very notable. To the best of my knowledge, the proposed DMFT approach (in this context) is novel and I find it to be very interesting. While it does not seem to be discussed in the paper in detail, I hope that this theoretical approach will have numerous other applications in the field. The theoretical approach itself being applied to a supervised problem (even though perhaps too simplistic) is of relevance for the NeurIPS community.

Weaknesses: One aspect of this work that I would consider somewhat weak is related to significance of the presented results. While the approach itself is very interesting, the considered toy models are very simplistic and it is not entirely clear whether this approach can lead to other insights, especially in more complex problems and settings. The derived system of integro-differential equations is quite non-trivial, difficult to solve computationally and it does not appear to provide many significant insights without approaching it numerically. Furthermore, many presented observations (dependence of generalization error on b, \lambda, etc) can be recovered by simply running SGD for sufficiently large finite d. In conclusion, this work provides a very interesting infinite-d model of SGD, but it is not clear if this leads to any improvements in our understanding of the SGD behavior and practical / future-work implications.

Correctness: I have not checked all derivations carefully, but the overall approach appears to be correct. Numerical experiments seem to support their correctness. Empirical methodology appears to be correct as well.

Clarity: The paper is overall well-written, but some derivations and results in the main text are not entirely clear and some symbols lack proper definitions. It is understandable that complex involved derivations are moved into Appendix, but the main conclusions/results should still be presented in the main text with all required definitions and explanations. For example, on line 136 the authors write "The limit d\to \infty is therefore controlled by a saddle point", but in my opinion, this phrase is not entirely clear in the context in isolation. The text is challenging to read because many of the symbols in Eq. (15) become clear only after looking into Appendix or going 1-2 pages forward into the main text. One quantity, R(t, t') (line 169) is in my opinion confusing even after reading Appendix since the text fails to explain what "infinitesimal local field H_i(t)" is and how it enters the dynamical equations (I could not find it anywhere in the text, or appendix).

Relation to Prior Work: The authors provide relevant prior work and clearly explain the novelty of their contributions. I would be curious to also see a discussion about the relation to the line of work on neural tanget kernel and some (maybe very brief) discussion of mean-field theories of other aspects of deep learning (like the work on the mean-field theory of batch normalization) if the authors judge it to be sufficiently relevant.

Reproducibility: Yes

Additional Feedback:


Review 2

Summary and Contributions: This work analyzes the SGD dynamics by using the dynamical mean field theory (DMFT) developed in the statistical physics of spin glass and learning. It gives the dynamics of generalization error in classification tasks of Gaussian mixtures with a linear model or a single-layer neural network model. By numerically solving the obtained equations, we can understand how the generalization performance depends on some control parameters.

Strengths: - DMFT enables us to reduce the high dimensional learning dynamics, which is too complicated to analyze, to some self-consistent equations characterized by just several parameters. - By numerically solving the obtained equations, we can see the dependence of the generalization error on various settings (e.g., batch size, regularization, initialization and input structure). In particular, it clarifies that the advantage of smaller batch size appears as a short of the early-stopping minimum of the generalization error at lager times.

Weaknesses: Although I am quite interested in using DMFT to explore modern machine-learning problems, I am not entirely convinced of the significance of the obtained results. - While I agree that the current DMFT formulation is novel compared to the conventional DMFT (in particular, that of [25]), my impression is that the adding finite-batch effect (Eq. (9)) is a bit straightforward and the technical progress from [25] is not so large. - If the insight obtained by the theory is significant enough, it will be not so important whether the technical progress is major or minor. However, I am unconvinced that the current work could give any suggestion or insight into major problems regarding the batch size. For example, one of the common beliefs on the advantage of SGD is that the noise caused by finite batch size helps the dynamics avoid poor local minima (e.g. non-zero training loss or sharp minima). It enables the dynamics to converge to better global minima with high generalization performance. Some studies empirically observed that specific optimal batch size, which is not too large but not too small, achieves the best generalization performance. It is unclear whether the DMFT analysis shown here is useful to analyze such well-known empirical observations of SGD. I will discuss more details in the additional feedback section.

Correctness: The derivation of the DMFT and empirical experiments are well described and hardly controversial.

Clarity: Most of the results are clearly presented, but the suboptimality of the dynamics in three-cluster case is unclear. I will discuss its details in the additional feedback section.

Relation to Prior Work: Relation to prior work is clearly discussed. This work differs from the DMFT of the previous work [25] in the following points; the current work extends the DMFT into the case fo finite batch size and structured input of Gaussian mixtures. It also differs from [21] which has analyzed the eventual generalization performance in a static way in the two-cluster case.

Reproducibility: Yes

Additional Feedback: - Two-cluster case is a convex optimization of the linear model and has been investigated in a bit different context [21]. Therefore, the three cluster case is more untrivial and exciting. However, I am not sure that the DMFT formulation in the three-cluster case is tractable enough to analyze SGD dynamics' behavior. Since the three-cluster case is non-convex optimization, I suspect that DMFT equations (20) have some local optima. If this is the case, it becomes unclear how typical the dynamics shown in experiments on three-cluster cases are. It is essential to answer how the batch size changes the local minima eventually selected by the SGD dynamics and the generalization performance of these minima. - [Smith & Li, A Bayesian Perspective on Generalization and Stochastic Gradient Descent, ICLR 2018] and series of their works observed that when the learning rate is properly scaled to the batch size, SGD has an optimal batch size to achieve the best generalization. Other works claim that the noise of SGD caused by finite batch size has anisotropic noise, and it is crucial to find such better minima (e.g. [Zhu et al., The Anisotropic Noise in Stochastic Gradient Descent: Its Behavior of Escaping from Sharp Minima and Regularization Effects, ICML 2019]). If the DMFT gives any quantitative justification of such empirical observations, it will increase the significance of this paper.


Review 3

Summary and Contributions: The authors provide a theoretical analysis of the training dynamics of Gaussian mixture classification. Unlike previous works studying in the one-pass (online) setting, in this work, the training data are reused multiple times.

Strengths: Studying the multi-pass data setting is generally hard. This work derived a set of closed equations for the training dynamics of the Gaussian mixture model. Though the model is simple, this work led a good initial step to study this reused-data setting.

Weaknesses: Though the authors claimed that they analyzed the dynamics of stochastic gradient descent (SGD), the batch size is O(n), rather than O(1). Thus, the randomness in the SGD may disappear in the large n limit. The authors may add some discussion on the setting of O(1) batch size, which may be more challenging.

Correctness: Yes

Clarity: Yes

Relation to Prior Work: Yes

Reproducibility: Yes

Additional Feedback: It’s a good work to study the training dynamics with reused data. The authors may provide some discussion on potential application/limitation of the dynamical mean-field theory on more complicated neural network models, e.g. is it possible to the same approach to analyze multi-layer networks in the recycled data scenario.


Review 4

Summary and Contributions: The authors study classification of structured datasets with a perceptron by considering an asymptotic limit of large dataset size and input dimension. In this limit the dynamics are reduced to those of a scalar stochastic process and some associated kernels, whose dynamics are solved for numerically in a self consistent manner to obtain the training and test error as functions of time, showing reasonable agreement with experiments. The authors use the tools of dynamical mean field theory that was developed to study disordered systems that are not in thermodynamic equilibrium. They extend previous work that considered unstructured data by including simple structure and considering a variant of gradient descent that uses a certain fraction of the data at every time point. I thank the authors for their rebuttal. Having read it and the other reviews, I still believe the paper should be accepted.

Strengths: Improving our understanding of the training dynamics of neural network is one of the major goals of theoretical research in machine learning. Some of these learning problems are naturally amenable to treatment with tools from statistical mechanics of disordered systems. Results on asymptotic generalization error in perceptrons for simple structured datasets have been known for several decades, yet the calculation of training and test errors as a function of time is a more recent development. The main novelty of the work is in extension of previous work to include the effects of a finite batch size and simple structured datasets. Since the batch choice can lead to a tradeoff between performance and the ability to parallelize training, and the optimal way to choose it is still not very well understood, improved theoretical understanding of its effect of training even in simple settings should be of interest to the community. The theoretical results of the paper are compared to experiment, showing reasonable agreement. The sources of some of the discrepancies are discussed.

Weaknesses: The main weakness is that it applies to simple models and datasets. While analyzing the dynamics of deep models is a distant goal, there are many results in the general vicinity of this one for two layer networks. The three cluster dataset can only be fit by the model used because of the non-standard choice of the nonlinearity. This indeed introduces nonconvexity in the form of the sign ambiguity as the authors note, but one wonders whether the dynamics will be indicative in any way of those in deeper models where the nonconvexity is a result of the nonlinearity and permutation invariance, while the output is generally not invariant to a sign change in the weights.

Correctness: The method is a standard one in statistical physics and the claims appear to be sound. There are some details in the experiments that are not provided (for instance about the process of solving the DMFT equations numerically).

Clarity: Overall, the presentation is clear. A few points where it can be improved are noted below: - The notation is slightly confusing in places - bold fonts are used for elements of vectors and superscripts with brackets are used to denote both time indices and elements of a vector in the same expression (in eq. A.1 for example). - In some places when an approximation is used yet this is written as an equality (for example the first equality in B.6).

Relation to Prior Work: Previous related work is discussed, and the novelty of this work---namely that the results apply to finite training times, structured data without requiring a one-pass assumption---are clearly stated.

Reproducibility: Yes

Additional Feedback: In figure 3, the minimum of the generalization error appears to depend in a non-monotonic way on the system size. Naively one would expect the dataset and model size to not make the problem any harder (since the model's structure is unchanged it should not be harder to optimize). How do the authors account for this? Did the authors compare the dynamics of persistent-SGD with standard SGD? There has been some interest in recent years in the relationship between the batch size and learning rate used in training neural networks, (for example [1] and references therein). In light of these, it might be interesting to explore the relationship between the discretization time scale and the parameter b in this setting as well. [1] Hoffer, Elad, Itay Hubara, and Daniel Soudry. "Train longer, generalize better: closing the generalization gap in large batch training of neural networks." Advances in Neural Information Processing Systems. 2017.

[Author Response · NeurIPS 2020]

We thank the reviewers (**R1**, **R2**, **R3**, **R4**) for their time and expertise. We are grateful to the reviewers for letting us
know which definitions and explanations were unclear, we will improve them in the revised version. We also thank
them for pointing out the typos, that we will correct in the revised version.
**R1, R2: Significance of the work –** We underline that our work is methodological in nature. Indeed, the introduction
of the finite batch size provides a conceptual step forward the understanding of the dynamics of mini-batch gradient
descent with DMFT. Moreover, in contrast to [25], considering structured data allows to investigate supervised learning
and questions on generalization properties. The theory we build opens way to many detailed investigations about the
nature of the difference between GD and SGD in simple models, and we are certain it will be followed by numerous
works (for examples of ongoing follow-up investigations, see below).
**R1, R2: Insights from theory and future works implications – (i)** DMFT provides a natural tool to characterize the
noise introduced by the finite batch size: at variance with [25], in Eq. (15) we have a Gaussian noise but also a stochastic
variable $s(t)$ which encodes the batch size noise. Recent literature (*arXiv:1912.00018*) investigated the behavior of SGD
gradient as a function of time and measured the corresponding statistical properties. Due to the dimensional reduction
performed by DMFT, comparing these experimental results with simple models by sampling the effective stochastic
process $h(t)$ is much easier than by direct simulations. In addition, we have direct access to two-point correlation
functions of the stochastic gradient which we expect to be present and important. Those are numerically much more
demanding to get in simulations than say the test error. The DMFT approach can instead be used to compute the full
time dependent correlations of the SGD gradient. Furthermore, from the response and correlation functions we can
extract an effective temperature (*arXiv:cond-mat/9611044*) which is a direct and quantitative measure of the noise that
governs the dynamics. In contrast, computing two-point response functions from numerical simulations is much more
challenging. **(ii)** Optimal stopping time: we are working on deriving an analytic formula for it as a function of the
model parameters.
**R1: Extensions to more realistic data –** We are working on generalizing the DMFT analysis of SGD to: **(i)** Models
of structured data with low intrinsic dimension embedded in large dimension, such as the Hidden Manifold Model [9],
or the Random Features Model (*arXiv:1908.05355*); **(ii)** Generalised Linear Models. DMFT can be used to study both
the recovery transition of Gradient Flow as well as how it changes when the finite batch size is employed.
**R1: Clarity of** $R(t, t')$ **–** Since we focus on the linear response regime, we couple an infinitesimal local field $H_i(t)$ to
each variable $w_i$, changing the loss as follows: $\mathcal{H}(\mathbf{w}) \to \mathcal{H}(\mathbf{w}) - \sum_{i=1}^{d} H_i w_i$, and hence adding and extra term $H_i(t)$
to the right hand side of Eq. (10). We will add a more detailed explanation in the revised version.
**R1: Other mean-field methods (NTK, mean field theory of batch normalization) –** We will extend the discussion
about mean-field analysis of infinitely wide networks in the introduction.
**R2, R4: Learning rate –** In our approach the learning rate is - strictly speaking - zero (we are in the stochastic gradient
flow regime) and enters only in the discretization of the DMFT equations for the numerical solutions.
**R2: "DMFT equations (20) have some local optima" –** This does not happen. The DMFT fixed point equations are
deterministic, hence at given initial condition the solution is unique. This can be seen as follows. The kernels computed
by DMFT are causal and a simple integration scheme of the equations is just extending those kernels progressively
in time starting just from their initial value which is completely deterministic given by the initial condition for the
stochastic process. Furthermore, we check this independently by simulations and we find a very good agreement with
the theoretical results even when the problem is non-convex. We will add this discussion.
**R2, R4: Optimal batch size –** We thank for this pointer, our methodology is indeed well fitted to address this question.
An important parameter in our setting is the persistence time, in Fig. 3 right we see that the smaller the persistence time
the better the early stopping error. We will investigate whether in other cases an optimum can be found.
**R3: The batch size is** $O(n)$ **and its noise averages out–** Even though we consider extensive batch size, the randomness
due to that does not disappear in the large $n$ limit. Indeed in Eq. (15) we have a Gaussian noise but also a stochastic
variable $s(t)$ encoding the batch size noise. Therefore even if the batch size is extensive, the resulting noise does not
average out. This is apparent e.g. in Fig. 1 left where the early stopping error depends clearly on the batch size.
**R3, R4: Extensions to two-layer architectures –** We are working on generalizing the DMFT to a committee machine
with finite number $(K)$ of hidden units (see for instance *arXiv:1806.05451*). In this case, instead of one effective
stochastic process for the typical gap as in Eq. (15), we will have $(K)$ coupled such processes.
**R4: Non-standard nonlinearity –** In relation to the point above, we can consider a committee machine, and the hidden
units together with the nonlinear activation will indeed introduce non-convexity. However, the fact that our analysis
allows to consider generic non-convex loss functions is interesting per se, as it is not the case for other methods in
existing literature.
**R4: Non monotonicity of generalization minimum in experiments –** In Figure 3, the exact location of the general-
ization minimum at finite dimension is not precise due to large fluctuations (see the errorbars).
**R4: Persistent-SGD vs SGD –** We compare the two algorithms in Figure 4.
**R4: Solving DMFT numerically –** We will add more details on that in the revised version.

[Meta-Review · NeurIPS 2020]

The reviewers agree that the techniques leveraged in the paper should be of interest to the wider NeurIPS community. Furthermore, even though the setting analyzed is relatively simple, the analysis is challenging, and understanding the effects of batch size is a problem of broad interest.